# Exercise Motives of College Students after the COVID-19 Lockdown

**DOI:** 10.3390/ijerph19126977

**Published:** 2022-06-07

**Authors:** Vojko Vučković, Klemen Krejač, Tanja Kajtna

**Affiliations:** Faculty of Sport, University of Ljubljana, 1000 Ljubljana, Slovenia; vojko.vuckovic@fsp.uni-lj.si (V.V.); klemen.krejac@fsp.uni-lj.si (K.K.)

**Keywords:** exercise motivation, EMI-2, physical activity, COVID-19 lockdown

## Abstract

This study describes the physical activity of university students (PA) and their motives to exercise in the summer of 2021, after the COVID-19 lockdown in Slovenia. Adults over 18 years of age (*n* = 493; 72% women) completed the International Physical Activity Questionnaire (IPAQ) short form and the Exercise Motivation Inventory-2 (EMI-2) via an anonymous online survey. Since the EMI-2 has never been used with the Slovenian population, the measurement properties of the Slovenian version (EMI-2) were also determined in this study. A forward–backward translation was used for cross-cultural adaptation. The internal consistency of the EMI-2 subscales was high. The results of the study showed that male students spent more minutes per session on intense physical activity and performed this activity more frequently per week, whereas female students were more likely to walk for more than 10 min without a break. In addition, female participants were more likely than male participants to cite weight management as a motivator. Male participants were more likely than female participants to cite fun, challenge, social recognition, belonging, competition, and strength and endurance as motivations. Results showed that respondents with a history of competitive sports had higher scores for revitalization, fun, challenge, social recognition, affiliation, competition, positive health, appearance, strength and endurance, and flexibility. They also engaged in intense physical activity more frequently per week, and when they engaged in moderate or intense PA activity, they did so for longer periods of time. Compared to respondents who never exercised, more of them overcame COVID-19. The results also showed some correlation between motivation and physical activity. The motives of revitalization, enjoyment, challenge, competition, strength and endurance, and stress management were more important for individuals who exercised vigorously more often in the past 7 days. Total time spent in daily physical activity is also related to the enjoyment of exercise and challenge. In conclusion, understanding the motives for exercise is important for the behavior of PA, especially after a period of inactivity due to lockdown.

## 1. Introduction

Since its outbreak, the pandemic triggered by COVID-19 has posed major physical and mental health challenges to populations worldwide [1,2]. One of the most important of these is physical inactivity [3,4,5,6,7,8,9,10,11,12,13,14,15], which can lead to physical [16] and mental risks [17,18]. Physical activity (PA) is defined as any physical movement produced by skeletal muscles that results in energy expenditure. This includes a wide range of activities such as walking, cycling, gardening, sports, and more [19]. Lots of studies have shown a decrease in PA in times of lockdown [20,21,22,23]. On the contrary, some studies have shown an increase in PA during lockdown [24,25], or after reopening [26]. In Slovenia, children and adolescents were less physically active during periods of lockdown. In 2021, there were 34.4% more obese children than in 2019 and their motor skills were 40% worse [27]. Slovenian adults who had exercised before lockdown were also less physically active during the lockdown and had more depression and anxiety symptoms [28].

Why are some people active and others not? It is known that lack of motivation is known to be one of the main reasons for physical inactivity [29,30]. Scientists agree that understanding the motives of individuals to choose a sports activity is of great practical value [31,32,33]. Some authors divide exercise motives into intrinsic and extrinsic. Intrinsic motives include motives such as enjoyment of exercise, feeling competent, and revitalization. Such motives have a positive effect on exercise endurance [34]. External motives are those that drive a person to exercise, such as the social status and recognition they enjoy from others, praise, or physical appearance. Such motives can also lead to anxiety, which often causes the person to stop exercising [32,35]. It is also known that exercise guided by internal motives has a positive effect on stress reduction. Conversely, training guided by external motives can lead to stress and even depression [32,36].

One of the more psychometrically valid motivational scales is the Exercise Motivation Inventory-2 [34]. Across the globe, studies have used the EMI-2 scale to measure exercise motives [37,38,39] in university students [40,41].

Some studies have shown that all items in the weight management subscale of EMI-2 targeted female participants, while all items in the competition subscale targeted male participants [42,43,44]. In contrast, Brazilian college students gave significantly greater importance to exercise for disease prevention reasons [45]. Similarly, in a study conducted among Filipino adolescents (15 to 21 years old), competition and challenge were the least frequently cited reasons for engaging in sports. The three most frequently cited reasons for exercising were: weight management, strength and endurance, and appearance [46]. How are motives connected with the amount of PA and exercise?

Some motives may be predictors of PA, such as stress management, enjoyment, competition, and weight management, while appearance may be a negative predictor for women [47]. In addition, motives such as enjoyment, affiliation, revitalization, and challenge are important in promoting intrinsic motivation to participate in sports by supporting the satisfaction of autonomy needs, which may positively affect the frequency of high-intensity exercise [48]. However, motivation for exercise based on appearance is significantly associated with increased rates of non-suicidal self-injury, particularly among high school and university students [49]. College students are particularly at risk for unhealthy exercise and diet, even during non-pandemic periods, and even more so when they are away from home [50]. It is known that female students exercise even less than male students [42]. Especially after the COVID-19 lockdown, it is known that lower physical activity is an important contributor to students’ poor well-being [51]. It is crucial to understand why PA habits of students are declining and also what is the motivation of those students that are regularly physically active. Therefore, this study was exploratory to determine if and to what extent students’ exercise motivation differs after COVID-19 lockdown—the aim of the present study was to determine the physical activity motives of university students in relation to selected sociodemographic indicators and physical activity levels. We hypothesized that higher intrinsic motivation would be related to (or predictive of) stronger physical activity behavior. The EMI-2 has never been used in the Slovenian area, so we will first translate the questionnaire back and forth and then validate it. We will calculate reliability using Cronbach’s alfa. Then, we wanted to investigate whether there was a relationship between the amount of PA and the motivational factors among Slovenian students in the summer of 2021, after the lockdown. We also wanted to find out if there are differences between participants who have participated in competitive sports in the past and participants who have never competed, as there is little literature on this topic.

## 2. Materials and Methods

### 2.1. Participants

The sample consisted of 493 subjects aged between 18 and 29 years, mean age was 21.69 years (±2.26 years). In total, 142 were men (*M* age = 21.65 ± 2.12 years) and 351 women (*M* age = 21.80 ± 2.59 years), the differences in age were not significant (*t* = −0.79; *sig* = 0.43). A total of 295 of them had previously exercised competitively (*M* age = 21.56 ± 2.15 years) and 198 had never played a sport in their lives (*M* age = 21.87 ± 2.36 years). The differences in age were not significant (*t* = 1.74; *sig* = 0.08). Inclusion criterion was that they had an active student status in the academic year 2020/21

### 2.2. Instruments

#### 2.2.1. EMI-2

The EMI-2 scale consists of 51 items and each item is measured on a 6-point Likert scale ranging from zero (does not apply to me at all) to five (applies to me very much), with higher scores indicating higher motivation to exercise. These items form up to 14 subscales, including: Affiliation, Appearance, Challenge, Competition, Fun, Health Pressure, Disease Prevention, Agility, Positive Health, Revitalization, Social Recognition, Strength and Endurance, Stress Management, and Weight Management. Each subscale is determined by calculating the average of 3 to 4 appropriate items based on the EMI-2 scale scoring key. The EMI-2 is a factorially valid means of assessing a wide range of motives for participation in sporting activities in adult men and women and is suitable for both athletes and non-athletes [34].

#### 2.2.2. IPAQ

The International Physical Activity Questionnaire (IPAQ) short questionnaire captures activity at four intensity levels: (1) vigorous activity such as aerobics, (2) moderate activity such as recreational cycling, (3) walking, and (4) sitting. The authors recommended the “last 7-day recall” version of the IPAQ-SF for physical activity monitoring studies, in part because the burden on participants to report their activity is low [52]. The IPAQ is a valid questionnaire in many EU countries [53] and also in the Slovenian population [54].

#### 2.2.3. Procedure

Study was conducted by approaching the students by e-mail through department chairs of each faculty of University of Ljubljana. Invitations were sent to all students, who were enrolled in the academic year 2020/21 and had an active student status and thus fulfilled the inclusion criterion. They were first asked to sign the informed consent and afterward they completed the questionnaire over an online platform, called 1ka (www.1ka.si, accessed on 15 May 2021). This research study was conducted in accordance with the Declaration of Helsinki and the Code of Ethics and Q4 Conduct of the British Psychological Society. The Ethics Committee of the Faculty of sport granted ethical approval for data collection, and all subjects provided written informed consent before participating in the study.

#### 2.2.4. Statistical Analysis

First, we conducted a factor analysis of the EMI-2 questionnaire and calculated the reliability of the extracted factors (using Cronbach’s Alpha). Then, we used these obtained factors to compare male and female students and to compare students who used to be involved in competitive sports and those who never competed—both comparisons were carried out using a *t*-test. We also checked the correlation between motivation and activity, which was carried out using the Pearson *r*.

## 3. Results

Based on the Figure 1 scree plot and analysis of eigenvalues, we decided to extract eight factors.

In Table 1 we can see how many components we can divide the questions of the questionnaire into. Most of the questions are in the first component, which accounts for 36.2% of the explained variance. Our factorization is not significantly different from the basic one, therefore, we will process the further results according to the scoring key introduced by Markland and Ingledew [34].

As we can see in Table 2, our results show that the internal consistency of the original EMI-2 scale is high for Slovenian students.

In Table 3 we can see the differences between female and male participants. We can see that females walk continuously for 10 min on more days a week than males. In addition, female participants cited weight management as motivation more often than male participants. Men, on the other hand, performed vigorous physical activity more frequently and invest more time in it overall than women. Male participants stated their motives as enjoyment, challenge, social recognition, affiliation, competition, and strength and endurance as motives.

In Table 4, we see a comparison between respondents who used to play competitive sports and those who did not. Respondents who used to play competitive sports had higher scores on revitalization, enjoyment, challenge, social recognition, affiliation, competition, positive health, appearance, strength and endurance, and nimbleness. They also engaged in intense physical activity more frequently per week, and when they engaged in moderate or intense PA activity, they did so for longer periods of time. Compared to respondents who never exercised, more of them overcame COVID-19.

Table 5 shows the correlation between motivation and physical activity. The motives revitalization, enjoyment, challenge, competition, strength and endurance, and stress management are more important for those who exercised more frequently in the last 7 days. All correlations are positive and moderate. Total time of daily intense physical activity is also related to the enjoyment of exercise and challenge. These associations are statistically significant at a rate of *p* < 0.01.

## 4. Discussion

As this was the first use of the EMI-2 questionnaire in this country, we first performed a factorization of the questionnaire and found that the obtained factors were very similar to the ones proposed by the authors of the questionnaire [34]. The comparison of male and female participants shows that females walk more than males when it comes to how many days per week they walk continuously for at least 10 min and they are more motivated by weight management. Males are more vigorously active and invest more time in physical activity and are motivated by enjoyment, challenge, social recognition, affiliation, competition, and strength and endurance. Differences between previous competitors and non-competitors showed that previous competitors were more motivated by different motives than non-competitors. Previous competitors also engaged in intense physical activity more frequently and for longer periods of time. We also found several correlations between motives for physical activity and actual physical activity and we will try to elaborate on these results in the discussion.

Although the benefits of regular exercise for physical and mental health are well known, only a small percentage of the population participates in this health behavior [2]. Because of the COVID-19 lockdown, people in Slovenia are even less active [28]. Consequently, this lack of regular exercise during the COVID-19 pandemic has led to an abundance of obesity-related diseases [16]. Many people who begin an exercise program drop out within the first 6 months [55]. Therefore, in order to understand what motivates people to engage in exercise, it is important to examine their motivations for participating in the exercise. 

We were the first researchers to use the EMI-2 in the geographical area of Slovenia. Our results show that the original scale has high reliability for Slovenian students; of the 14 motivational subscales, 13 have a Cronbach’s Alpha greater than 0.8 and one is greater than 0.7. For this reason, we did not feel it was appropriate to change anything and used the original scale and we can say that this questionnaire is appropriate for use in this country; verifying this was namely the first aim of this research.

Our study in Slovenia, similar to other studies, showed that male students spent more minutes per session in intense physical activity than females [42]. The results of our study also suggest that male students engage in intense physical activity more often per week. Some other authors also indicated that total physical activity scores were higher in male participants than in female participants [47]. In contrast, our results suggest that female students are more likely to walk for more than 10 min without rest. We could say that males are more active when it comes to vigorous physical activity, but not more active in all types of physical activity.

This study also confirmed previous findings that males and females have different motivations to engage in physical activity. Female participants in our study reported more weight management motivations for participating in exercise and physical activity, similar to several other studies [40,41,42,43,56]. Some other studies suggest that females are also more motivated by appearance [40,42,47], positive health and stress management [42,47], and sometimes exercise [42]; however, our results did not confirm this.

Our results suggest that men are more often motivated by fun than women, which was also shown by [47]. Males reported higher levels of challenge motivation than females, similar to the studies by Pauline [42] and Ednie and Stibor [47].

In our study, male participants rated the social recognition motive higher than females on average. This result is consistent with many other studies [42,43,47,49]. Our results also suggest that motives related to affiliation are stronger in male participants, which was also demonstrated by Pauline [42]. Strength and endurance motives are also predominantly associated with male study participants, as found in several other studies [42,43]. We found that the strongest association in males was competition motives. The importance of competition motives for men was described in several other studies [42,43,47,49,56], but in contrast, studies in the Philippines [46] and Brazil [45] found opposite results, namely that competition was among the least important reason for exercising. 

As we can see, men were more often motivated by performance and ego-oriented factors, while women were more often motivated by weight management factors. Some authors suggest that some motives such as affiliation, challenge, and motivation are significantly higher among participants in competitive physical activities [56]. It is also known that competition is positively associated with higher levels of intrinsic motivation [57]. To the best of our knowledge, our study is the first to use the EMI-2 to examine whether there is a difference in motivation between participants who have exercised competitively in the past and participants who have never exercised. Thus we are unable to compare our result with previous studies, but this is an area to be explored further—we found that ex-competitors are more active and as being active is a health-maintaining behavior, this deserves a bit more scientific attention. It was previously shown that ex-competitors, especially after the COVID-19 lockdown, are more immune to some mood disorders such as depression and anxiety [19].

In more detail, our results show that ex-competitors were significantly more active. They were physically active more often and longer per session than the participants who never exercised. They also had more moderate PA per day. Our results also suggest that ex-competitors’ and noncompetitors’ ratings of motivation differed for 11 of the 14 motives. Ex-competitors rated revitalization, enjoyment, challenge, social recognition, affiliation, competition, positive health, appearance, strength and endurance, nimbleness, and stress management significantly higher than noncompetitors.

Some studies [47] suggest that participants with higher scores on the stress management (men) and revitalization, weight management, and fitness (women) scales had higher PA total scores after controlling for other variables. As expected, our results suggest that there is a moderate but significant relationship between the frequency of intense PA and the motives of revitalization, pleasure, challenge, competition, strength and endurance, and stress management. The motives of fun and challenge are also positively related to the amount of daily physical activity PA. College students who have the above motives may exercise more.

We tried to explore whether there might be a relationship between physical activity and symptom severity of COVID-19, but we found nothing.

### Limitations

This study is subject to several limitations. First, this sample is representative of the University of Ljubljana, which is located in central Slovenia and may not be representative of other geographic regions.

Second, the sampling methodology was voluntary and self-selected. A limitation of this type of methodology is that the subjects are usually individuals who have strong opinions about the topic under study.

Third, the online questionnaire was conducted within a few weeks, so respondents did not answer it at the same time. In addition, the study was cross-sectional, providing only a snapshot of the current state. Understanding exercise motivation and how motivation may change over time for individuals would be better observed if a longitudinal study were conducted.

Fourth, the EMI-2 does not specifically distinguish between exercise and sport, so it was not possible to classify what type of “sport” (e.g., endurance sport or game sport) or “exercise” the students actually engaged in. This type of information would be useful in preparing future physical activity initiatives for university students and will therefore be included in a follow-up study.

Because we collected our sample (*n* = 494) via email and Facebook pages, it contains a high proportion of students enrolled in either sport- or health-related programs and may not be fully representative of a ‘typical’ Slovenian university student.

## 5. Conclusions

Our study shows, that the Slovenian version of the EMI-2 questionnaire shows high reliability and a factor structure very similar to the original one, which means that it can be used in this country to the same end as the original version, which is measuring motivation for physical activity. We compared physical activity and motivation for physical activity in male and female participants of the study and confirmed several differences. Females walk more than males (when comparing the number of days per week they walk continuously for at least 10 min), and women are also more motivated by weight management. Males are more vigorously active and invest more time in physical activity, their motivation for physical activity is enjoying themselves, looking for challenges and social recognition. This means that different approaches need to be used when approaching men and women to engage in more physical activity. We also searched for differences in motivation for physical activity and actual physical activity between previous competitors and non-competitors and found several. As there is a lack of research in this area, we would like to encourage more researchers to investigate the effect of a previous competitive career on future physical activity and the motivational structure for physical activity. Our results also confirm that students who are motivated by revitalization, challenge, pleasure and competition, are more physically active.

## Figures and Tables

**Figure 1 ijerph-19-06977-f001:**
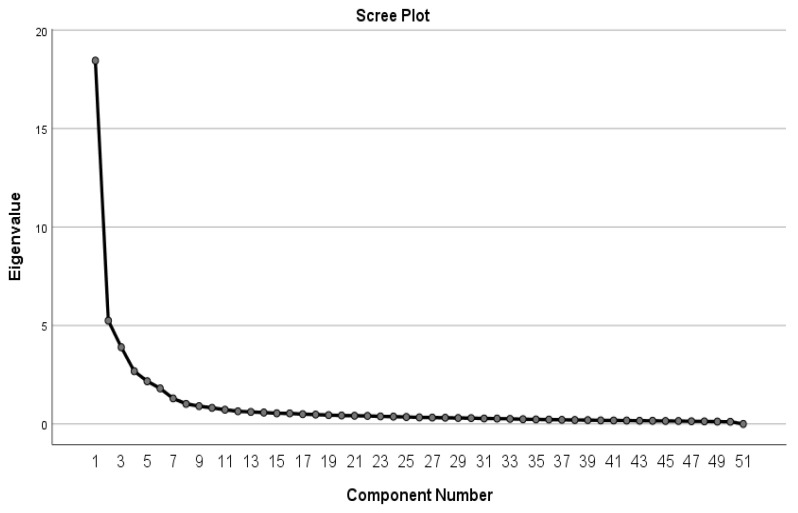
Scree plot of the factor analysis of the EMI-2 questionnaire.

**Table 1 ijerph-19-06977-t001:** Eigenvalues of the factors, extracted in factor analysis of the EMI-2 questionnaire.

Component	Initial Eigenvalues			Extraction Sums of Squared Loadings			Rotation Sums of Squared Loadings		
	Total	% of Variance	Cumulative %	Total	% of Variance	Cumulative %	Total	% of Variance	Cumulative %
1	18.45	36.17	36.17	18.45	36.17	36.17	8.24	16.15	16.15
2	5.24	10.28	46.46	5.24	10.28	46.46	6.54	12.81	28.96
3	3.89	7.63	54.09	3.89	7.63	54.09	4.41	8.65	37.62
4	2.68	5.25	59.34	2.68	5.25	59.34	4.27	8.38	45.99
5	2.17	4.25	63.60	2.17	4.25	63.60	4.21	8.26	54.25
6	1.80	3.54	67.13	1.80	3.54	67.13	3.35	6.56	60.81
7	1.30	2.54	69.67	1.30	2.54	69.67	3.05	5.99	66.80
8	1.02	2.00	71.67	1.02	2.00	71.67	2.48	4.87	71.67
9	0.90	1.77	73.44						

**Table 2 ijerph-19-06977-t002:** EMI-2 original scale reliability for Slovenian College students.

	Cronbach’s Alpha	N of Items
Stress Management	0.89	4
Revitalization	0.825	3
Enjoyment	0.91	4
Challenge	0.84	4
Social Recognition	0.84	4
Affiliation	0.91	4
Competition	0.93	4
Health Pressures	0.70	3
Ill Health Avoidance	0.81	3
Positive Health	0.88	3
Weight Management	0.87	4
Appearance	0.94	4
Strength and Endurance	0.91	4
Nimbleness	0.85	3

**Table 3 ijerph-19-06977-t003:** Gender differences between motives.

	Female			Male						95% CI for Cohen’s d	
	*N*	*M*	*SD*	*N*	*M*	*SD*	*t*	*sig*	*Cohen’s d*	*Lower*	*Upper*
Vigorous_7days	351	2.49	1.95	142	3.37	1.98	−4.56	0.00	−0.44	−0.64	−0.25
Vigorous_Minutes_Day	265	61.54	38.47	102	76.90	39.09	−3.41	0.00	−0.39	−0.62	−0.17
Walk_10minutes_7days	351	5.57	1.86	142	5.08	2.26	2.27	0.02	0.24	0.03	0.45
Enjoyment	351	12.63	5.61	142	14.14	5.61	−2.71	0.01	−0.27	−0.46	−0.07
Challeange	351	10.89	5.58	142	12.57	5.51	−3.04	0.00	−0.30	−0.49	−0.11
Social_Recognition	351	5.85	5.11	142	7.65	5.36	−3.50	0.00	−0.34	−0.54	−0.15
Affiliation	351	8.96	6.10	142	10.97	6.21	−3.30	0.00	−0.33	−0.52	−0.13
Competition	351	5.99	5.93	142	9.87	6.93	−6.25	0.00	−0.60	−0.80	−0.40
Weight_Management	351	12.33	5.40	142	10.34	6.11	3.56	0.00	0.35	0.15	0.55
Strength_Endurance	351	14.89	5.04	142	16.23	4.16	−2.79	0.01	−0.28	−0.46	−0.10

**Table 4 ijerph-19-06977-t004:** History of competing differences between motives.

	Not Competitors			Ex-Competitors						95% CI for Cohen’s d	
	*N*	*M*	*SD*	*N*	*M*	*SD*	*t*	*sig*	*Cohen’s d*	*Lower*	*Upper*
Vigorous_7days	198	2.44	2.02	295	2.94	1.95	−2.74	0.01	−0.25	−0.43	−0.07
Vigorous_Minutes_Day	146	54.03	36.77	221	73.59	38.88	−4.82	0.00	−0.50	−0.70	−0.30
Moderate_Minutes_Day	124	52.99	38.62	212	64.21	38.63	−2.57	0.01	−0.29	−0.51	−0.07
Revitalisation	198	9.84	3.98	295	11.61	3.15	−5.23	0.00	−0.49	−0.67	−0.30
Enjoyment	198	10.97	6.09	295	14.47	4.85	−6.77	0.00	−0.62	−0.80	−0.44
Challenge	198	9.58	5.68	295	12.58	5.24	−5.93	0.00	−0.53	−0.71	−0.36
Social_Recognition	198	4.77	4.61	295	7.44	5.37	−5.89	0.00	−0.51	−0.68	−0.34
Affiliation	198	7.51	5.87	295	10.90	6.04	−6.19	0.00	−0.55	−0.72	−0.38
Competition	198	4.03	4.89	295	9.18	6.59	−9.95	0.00	−0.80	−0.95	−0.64
Positive_Health	198	11.62	3.83	295	12.64	2.79	−3.21	0.00	−0.31	−0.50	−0.12
Appearance	198	12.92	5.94	295	14.13	5.13	−2.35	0.02	−0.22	−0.41	−0.04
Strength_Endurance	198	14.37	5.33	295	15.88	4.38	−3.30	0.00	−0.31	−0.50	−0.13
Nimbleness	198	9.99	3.99	295	10.72	3.58	−2.07	0.04	−0.19	−0.38	−0.01
COVID-19	198	1.25	0.44	295	1.34	0.47	−2.00	0.05	−0.18	−0.36	0.00
Stress_Management	198	11.36	5.68	295	13.80	4.87	−4.94	0.00	−0.46	−0.64	−0.27

**Table 5 ijerph-19-06977-t005:** Correlations between amount of activity and motives for recreational activity.

		Vigorous_7 days	Vigorous_Minute_Day
Revitalisation	*r*	0.32 **	
Enjoyment	*r*	0.41 **	0.32 **
Challenge	*r*	0.42 **	0.35 **
Competition	*r*	0.30 **	
Strength_Endurance	*r*	0.36 **	
Stress_Management	*r*	0.31 **	

** Correlation is significant at the 0.01 level (2-tailed).

## Data Availability

The data presented in this study are available on request from the corresponding author. The data are not publicly available due to the clinical nature of the used questionnaire subscales, which were approved specifically for this research.

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
