# Peer review of "Exercise Motives of College Students after the COVID-19 Lockdown"

_ijerph, 2022, doi:10.3390/ijerph19126977_

Round 1

Reviewer 1 Report

Thank you for giving me the opportunity to read and comment a report “Exercise motives of college students after the COVID-19 lockdown”, by Vuckovic V, et al.

This is a potentially interesting report but at present it is not suitable for publication

  • The introduction section is out of order. This section should end with a clear indication of the main aim of the study. In addition, it is recommended that the authors review the entire introduction section, as they repeat the same concepts several times.
  • In the Participants subsection, it would be advisable to remove the first sentence, since the aim of the study should be in the Introduction section.
  • It is necessary to include a section explaining the statistical analysis performed by the authors, otherwise it is difficult to understand the results obtained.
  • The acronym 1ka appears on line 52. It would be advisable for the authors to explain their meaning.
  • In the opinion of this reviewer, it is necessary to modify the titles of Figure 1, Table 1 and Table 5, as they are not self-explanatory.
  • It is necessary for the authors to review all the results, since in the Anglo-Saxon format decimals are expressed by points, not by commas. In addition, it would be convenient to unify the number of decimals used in the different tables.
  • Normally, the Discussion section begins with a summary of the main findings obtained by the authors.
  • Finally, it would be advisable to review the bibliography, since the references do not follow the format established by the journal.

Author Response

  • The introduction section is out of order. This section should end with a clear indication of the main aim of the study. In addition, it is recommended that the authors review the entire introduction section, as they repeat the same concepts several times.

Dear reviewer, thank you for the careful reading of the introduction. We have shortened the introduction section, erased the concepts that were repeated and rewrote ending of introduction section, so that the main aim of the study is clearer. We hope you will enjoy reading the improved version of the introduction.

  • In the Participants subsection, it would be advisable to remove the first sentence, since the aim of the study should be in the Introduction section.

This sentence was moved higher up into the introduction, where the aim of the study was explained in a more detailed manner.

  • It is necessary to include a section explaining the statistical analysis performed by the authors, otherwise it is difficult to understand the results obtained.

We added the section about statistical analysis, thank you for the recommendation.

  • The acronym 1ka appears on line 52. It would be advisable for the authors to explain their meaning.

The explanation was included.

  • In the opinion of this reviewer, it is necessary to modify the titles of Figure 1, Table 1 and Table 5, as they are not self-explanatory.

Titles of the figure and both suggested tables were corrected, we hope you will find these titles to be more suitable.

  • It is necessary for the authors to review all the results, since in the Anglo-Saxon format decimals are expressed by points, not by commas. In addition, it would be convenient to unify the number of decimals used in the different tables.

Thank you for this comment – we changed all commas in the tables into points and unified the number of decimals – now two decimals are shown in all the tables.

  • Normally, the Discussion section begins with a summary of the main findings obtained by the authors.

An introductory paragraph was added into the discussion, which describes the main findings, presented in the results section.

  • Finally, it would be advisable to review the bibliography, since the references do not follow the format established by the journal.

The bibliography was thoroughly revised, we hope we used the correct type of the format.

Reviewer 2 Report

 Thank you for the opportunity to review this nanuscript. 

I think that the overall structure and writing of introduction part are not clear and well-aligned because it is not easy to catch what the research questions and strategies in this paper. Please clearly describe those things. As you already knew, the introduction section is one of the most important parts not only to draw attention of readers but also to provide guidelines for them to facilitate a clear understanding of the paper. Parts of the introduction are more suitable for the Discussions section.

I recommend ading the section  - Study design.

The Participants section includes the purpose of the research and a few details about the topics. The inclusion and exclusion of participants must be detailed.

The Discussion section needs to be revised and extending to facilitate the understanding of the most relevant evidence identified in the study compared to previous studies.

The conclusions are only partially relevant in the context of the topic of the manuscript.

Author Response

I think that the overall structure and writing of introduction part are not clear and well-aligned because it is not easy to catch what the research questions and strategies in this paper. Please clearly describe those things. As you already knew, the introduction section is one of the most important parts not only to draw attention of readers but also to provide guidelines for them to facilitate a clear understanding of the paper. Parts of the introduction are more suitable for the Discussions section.

Dear reviewer, thank you for the careful reading of our article. We have shortened the introduction section, so that the readers understand the background, previous studies in the field of our research and the main aim of the study. We provided guidelines for readers and described research questions. We hope you will find our improvements suitable.

I recommend adding the section  - Study design.

We added all the necessary information to the section procedure, where the study design is now well described – we hope you will find these corrections suitable.

The Participants section includes the purpose of the research and a few details about the topics. The inclusion and exclusion of participants must be detailed.

The participants section was improved – we added detailed information of the age and significance of age differences between men and women and previously active competitors and participants, who never competed in sports. We also added the inclusion criteria for participation in the study.

The Discussion section needs to be revised and extending to facilitate the understanding of the most relevant evidence identified in the study compared to previous studies.

We reorganized the discussion and added an introductory paragraph, but relatively few changes were made from a simple reason – the other reviewers suggested that the discussion is very good and suggested not to change it. We decided that too many changes in the discussion might affect this opinion, we hope you will be satisfied with the amendments made.

The conclusions are only partially relevant in the context of the topic of the manuscript.

We rewrote the entire section, created a new, more suitable conclusion to the article.

Round 2

Reviewer 1 Report

The manuscript has been improved and is suitable for publication.

Reviewer 2 Report

The authors improved the manuscript according with the recommendations.